# The Contents and Functions of the 49-Day Funeral Rites in Modern Korean Buddhism

Hyungong Moon and Brian D. Somers *

Department of Buddhist Studies, Dongguk University, Seoul 04620, Republic of Korea; darkmhg@dongguk.edu
* Correspondence: briansomers29@dongguk.edu

**Abstract:** This article explores the history and procedures of the 49-day Buddhist funeral ceremony, which functions as a ritual for the dead and a healing tool for the bereaved. The significance of this ceremony has its origins in *The Treatise of the Great Commentary of the Abhidharma* (아비달마대비바사론, 阿毘達磨大毘婆沙論) and *The Sūtra of the Fundamental Vows of the Bodhisattva Kṣitigarbha* (지장보살본원경, 地藏菩薩本願經). While this 49-day ceremony has been practiced in Korea for centuries, it was overshadowed by Confucian-style funerals, which were predominant during the Joseon dynasty. Since the end of the Joseon dynasty, Buddhism and Buddhist practices, including the 49-day funeral ceremony, emerged in Korea and continue to be practiced with frequency today. This article maintains that these rituals have two primary purposes. The first is to aid the departed in a successful rebirth. The second is to help the bereaved cope with their loss, which often includes various forms of psychological distress. After introducing the 49-day ceremony as it is currently practiced in Korea, this article shifts its focus to delve deeper into the ceremony's potential for healing. We will first examine the psychological healing elements that this ceremony offers, followed by considerations related to the grieving process, both within and outside of a Buddhist context.

**Keywords:** Korean Buddhism; Buddhist funeral rites; 49-day ceremony; grief; bereavement; healing; Pure Land

## 1. Introduction

Among the many sufferings of human life, the loss of a loved one is one of the most painful. In addition to the expected deaths of the elderly, which are in themselves difficult, sudden deaths due to natural disasters, road accidents, fires, building collapses, and suicides can be exceptionally challenging. Regardless of the cause, death leaves the living with scars caused by grief and bereavement. Religious rituals are an effective means by which loss can be processed. Often, religious ceremonies offer a necessary way to send the deceased to a noble state or realm, allowing the bereaved an opportunity to process their grief (Lim 2016, p. 238). While one's grieving period varies depending on the individual and the circumstances of her bereavement, leave from work is customary and, depending on the country, it may be required by law. Three to seven days of bereavement leave is a general standard (Chiappetta 2022); however, this varies from culture to culture. While three to seven days may be a sufficient length of time to manage the arrangements associated with death, it has been suggested that the relatively short period of time required to prepare and lay to rest the remains and perform the associated rituals may be insufficient for the bereaved who are left with grief and shock long after the ceremony's completion (Koo 2008b, p. 298).

With sufficient time, the bereaved learn to accept the new life without their loved one. Feelings related to the loss, such as guilt, anger, longing, and the like, diminish in time, and survivors return to a functional state. While it is important for the bereaved to process their loss on their own, it is also important to have communal support. Therefore, while loss is a problem, how one copes with it for the weeks, months, and years following the

loss is very important. Those who experience it must be supported in a variety of ways so that they can have a full grieving process (Yu 2018, p. 3).

The 49-day Buddhist funeral ceremony (henceforth referred to as the 49-day ceremony) (사십구재, 四十九齋)[1] is a typical funeral ceremony in Korean Buddhism, which takes place over 49 days, providing bereaved families an ample opportunity to interact with each other and the spirit of the dead. This article will focus on two primary functions of the 49-day ceremony, which relate to the dead and the living, respectively. Firstly, the ceremony functions as a send-off for the deceased, ensuring their place in the Pure Land. Second, the ceremony acts as a means of spiritual and psychological comfort for the survivors. In addition, the 49-day ceremony also functions as a means of communal healing. For example, this ceremony was held in South Korea for the victims of the Itaewon tragedy of 2022, where 156 young people passed away (Park 2022; Lee 2022). It was also practiced in 2014 following the Sewol ferry disaster when 306 lost their lives, including 250 students (Joe 2014). Furthermore, in 2009, a 49-day ceremony was held for former Korean president Roh Moo-hyun, with many people participating nationwide. The former president's 49-day ceremony was particularly significant as it served as a backdrop for the growing public interest in the 49-day ceremony in South Korea (Koo 2010, pp. 270–71).

This article will explore the origins and Buddhist background, as well as a detailed look at the procedures involved in the 49-day ceremony. In particular, the procedures of the 49-day ceremony in the *Tongil beobyojip* (통일법요집, 統一法要集) (DPBJO 2003) of the Jogye Order of Korean Buddhism will be focused on. Before the *Tongil beobyojip* was published in 1998, *The Book of Buddhist Liturgies* (석문의범, 釋門儀範) was used, but the *Tongil beobyojip* is now the most used Buddhist ritual book in most temples.

The 49-day ceremony of Korean Buddhism has been passed down through the ancient Three Kingdoms period, the Goryeo Dynasty, the Joseon Dynasty, and the modern era (Lim 2021, p. 141). There have already been historical and literary studies on this topic; however, this article focuses on the 49-day ceremony as it is practiced in contemporary Korean society, particularly based on studies within the past twenty years. In particular, this article examines the functions of the 49-day ceremony pertaining to the guidance of the spirit of the dead and the psychological healing of the bereaved.

## 2. The Origin and History of the Korean Buddhist 49-Day Ceremony

The 49-day ceremony refers to the seven funeral ceremonies held once every seven days for a total of 49 days. The significance of 49 days comes from the Buddhist belief that the deceased spends 49 days passing from this world to the next. This appears in *The Treatise of the Great Commentary on the Abhidharma* (아비달마대비바사론, 阿毘達磨大毘婆沙論), compiled in the mid-second century (Lim 2021, p. 112). The śāstra states, 'The longest that the intermediate existence (중유, 中有) can remain is seven times seven days (칠칠일, 七七日), and on the 49th day one will surely be reborn'.[2] Intermediate existence refers to one of the four stages of the life of a reincarnated being, presented alongside birth existence (생유, 生有), basic existence: the period of living after birth (본유, 本有), and death existence (사유, 死有). Intermediate existence is the stage after death but before the next life (Koo 2009, p. 229). *The Treatise on the Foundation for Yoga Practitioners* (유가사지론, 瑜伽師地論) mentions intermediate existence as follows:

> If this intermediate existence has not yet acquired the conditions for the next life, it will stay for seven days, but even if it does acquire the conditions for birth, the date of birth is not immediately determined. If by the seventh day it has not yet acquired the conditions to be born, it dies and is reborn and stays for seven days. Wandering in this way, it stays until the $7 \times 7$th (49 days), and from then on, it will surely acquire the condition that must be born.[3]

According to this quotation, during the stage of intermediary existence, if the ghost of the deceased does not meet the conditions to receive the next life within seven days, it dies and is reborn within the same realm. This process may be repeated up to seven

times, ultimately resulting in one receiving their next life on the 49th day.[4] *The Sūtra of the Fundamental Vows of the Bodhisattva Kṣitigarbha* (지장보살본원경, 地藏菩薩本願經) makes the following additional remarks:

> If one does many good deeds during the 49 days after the death of the body, all sentient beings will abandon unfortunate destinies (惡趣, unfortunate rebirth; three evil destinies; hells, a hungry ghost, an animal) and receive great pleasure by being born in the human or heavenly realms. The benefit to the living family will be infinite.[5]

This sūtra supports the significance of the 49-day ceremony by stating that when the bereaved perform rituals for the deceased, they create good karma not only for the deceased but for themselves as well. This posthumous ritual is premised on reincarnation and is, therefore, centered on sending the dead to a better afterlife (Koo 2009, pp. 247–48) but also accounts for the lives of the bereaved. The significance is that rituals for the dead affect the living.

The 49-day ceremony has had a long history in Korea. The basic route that Buddhism took when coming to Korea was from India through China. Therefore, it is speculated that the Korean Buddhist view of death, which was formed during the adoption of Buddhism in Korea, is a mixture of three views. The first is the Indian Buddhist view of life and death; the second is the Indian Buddhist view of life and death combined with Chinese philosophy, and the third is the Chinese Buddhist view of life and death combined with Korean folk beliefs (Gu 2020, p. 162). It is said that the Mahayana Buddhist view of the afterlife was introduced to Korea in earnest after the compilation of the *Bulseol yesu siwangsaeng chilgyeong* or *The Scripture Spoken by the Buddha on Preparing the Ten Kings Ritual for Rebirth after Seven Days* (불설예수시왕생칠경, 佛說預修十王生七經) at the end of the Tang Dynasty in China. The ten kings in the sutra's title refer to ten judges who weigh the merits of a person's karma in the afterlife, a concept that has its origins in Chinese Taoist thought (Gu 2020, p. 167). Although much more is known about funeral rites and practices regarding the 49-day ceremony during Tang and Song China, it seems that it was much more commonly practiced in medieval Korea than was originally thought (Vermeersch 2014, p. 46). Kwon (2019) considers evidence that belief, in Korea, in a Buddhist intermediary existence and the ten kings of hell was practiced as early as the Goryeo Dynasty (918–1392 CE).

Furthermore, in *The Great Vows of Kṣitigarbha Bodhisattva*, Kṣitigarbha is described as rescuing sentient beings from hell. Moreover, in the *Bulseol yesu siwangsaeng chilgyeong*, the king of the world of the dead (염라대왕, 閻羅大王), the ruler of the ten kings, is described as having the same role as Kṣitigarbha bodhisattva (Ra 2005, p. 139). The 49-day ceremony recognizes Kṣitigarbha, which stems from this historical background. Thus, the 49-day ceremony can be thought of as a fusion of many aspects of history and culture, including Mahayana Buddhist thought and indigenous Chinese beliefs.

Unlike Mahayana Buddhism, which came to Korea through China, Theravadin Buddhism has a somewhat different view of death. In the *Abhidhammattha Saṅgaha*, which epitomizes the Theravada Buddhist view of death, at the end of one's life, the marks (*nimitta*) of his/her karma are revealed, indicating the next life he/she will be reborn into. Upon reaching the end of one's life, the mind at the time of death (*cuti citta*) arises and is immediately connected with the consciousness that connects this life with the next (*paṭisandhi*) (Narada 1979, pp. 301–4). Therefore, there is no intermediate state between this life and the next, like there is in Mahayana Buddhism (Lim 2021, p. 138). These differences are reflected in the funeral rituals. Theravada Buddhism does not have funeral rituals like the 49-day ceremony (Lim 2021, p. 139). In addition, the funeral rituals of Thailand and Sri Lanka, two countries representative of Theravadin Buddhism, do not appear to have funeral rituals centered on *Tongil beobyojip*. Their predominant view of death and rebirth can be described as a kind of 'no gap reincarnation theory', and thus, the rituals associated with the 49-day ceremony do not exist (Lim 2021, pp. 145–50). Buddhist funeral rituals vary according to historical processes and cultural backgrounds, even among Buddhist countries.

The 49-day ceremony, as practiced in Korea, although influenced by early Buddhism, is based on Mahayana traditions.

While some scholars claim that the 49-day ceremony was practiced during the Unified Silla and Goryeo periods, this issue is debated. This ambiguity is even more unclear, given that the 49-day ceremony has also been referred to as the seven–seven-day ceremony (칠칠재, 七七齋), which is said to have begun in the middle of the Goryeo Dynasty (Na 2008, p. 263). There are no Korean sources from the Unified Silla and Goryeo periods that contain either of the terms, the 49-day ceremony or the seven–seven-day ceremony. However, in the Veritable Records of the Joseon Dynasty (조선왕조실록, 朝鮮王朝實錄), a comprehensive history book that records the deeds of the kings of the Joseon Dynasty, the term seven–seven-day ceremony repeatedly appears. Some examples include, 'At Jingwansa Temple (津寬寺), Queen Daehaeng's (大行 王妃) seven–seven-day ceremony was held.' (設大行王妃七齋于津寬寺) (JWS 2023a) and 'When a person dies, the seven–seven-day ceremony is performed to lead him on a good path…' (人死則皆欲薦拔, 旣設七七之齊,) (JWS 2023b). From these records, it is evident that the 49-day ceremony was observed during the Joseon Dynasty. One might deduce that these practices were established before the recorded instances, given that they did not emerge abruptly. However, the exact timeframe of the ceremony's inception preceding these records remains unclear.

The 49-day ceremony was commonly practiced by various classes, including monks, civilians, and the royal family (DPBJO 2003, p. l69). The Goryeo Dynasty was a Buddhist-centric state and Buddhist ideas and rituals deeply influenced its people, leading to the widespread adoption of Buddhist rituals. It was common for a departed spirit to be enshrined in a temple, and the rituals were performed accordingly (Koo 2008a, p. 239). During the Joseon Dynasty (1392–1897), however, Buddhism was sidelined throughout the entire period by the policy of *sungyu eokbul*, which means 'promoting Confucianism and rejecting Buddhism' (숭유억불, 崇儒抑佛). Although many temples were closed or turned over to the state, Buddhism, which had been rooted in the culture of the common people since the Goryeo Dynasty, remained in some rituals, including funeral rites (Han 2008, p. 138). In addition, the ritual space for funerals, which had been conducted under Goryeo Buddhist influence, gradually shifted from temples to homes (Koo 2008a, p. 239). Despite the active suppression of Buddhism and Buddhist practice, which had been largely replaced by Confucianism and Confucian norms, content related to the 49-day ceremony was recorded in *The Veritable Records of the Joseon Dynasty*, which has helped the tradition survive alongside Confucian funerals (Jung 2018, p. 34). It is speculated that Confucianism, which does not recognize reincarnation, did not satisfy the desire for life after death and that the Buddhist idea of reincarnation was needed in the royal family. In other words, regarding funeral practices during the Joseon Dynasty, Confucian funerals prevailed. However, depending on the religious aspiration for an afterlife, the 49-day ceremony seems to have been intermittently practiced (Koo 2008a, p. 239).

Joseon Dynasty Confucian funerary culture, along with traditional Korean shamanism, have contributed to the complex nature of the Korean 49-day ceremony. In Confucianism, filial piety (효, 孝) is highly valued and centers around respect for parents and ancestors, as well as ancestral rites (Gu 2020, p. 174), a feature held in common with the 49-day ceremony. In Korean Buddhist temples, there are dedicated spaces for the Buddha and bodhisattvas, as well as spaces for the spirit of the dead and ancestors. In this space, photos of the deceased are enshrined during the 49-day ceremony, which is especially important during the seventh stage, the performance of memorial rites (관음시식, 觀音施食), when food is offered to the ancestors. The integration of Confucian culture with Buddhism is evident in the dual usage of spaces devoted to Buddhas and bodhisattvas, as well as the devotion to ancestors and the deceased (Koo 2010, p. 267).

In addition to Confucian thought, the 49-day ceremony is also influenced by Korean shamanism (무속, 巫俗) and Korean folk religion. Here, the first character, *mu* (무, 巫), refers to the shaman (무당, 巫堂), who acts as a mediator between the dead and the living. While Korean shamanism is characterized by the worship of ancestors and indigenous

deities, it is particularly marked by the resolution of resentment (해원, 解冤). This means that through religious rituals, the dead resolve (解) their grievance or injustice (冤) during their lifetime and return to the afterlife (Koo 2010, p. 266).

Also, among the procedures in the 49-day ceremony is anointing and bathing of the spirit (관욕, 灌浴) to purify the deceased of bad karma. Among the Korean shamanic resentment resolution rituals are washing (씻기) and resolving (풀기), which are very similar to the bathing of the spirit in the 49-day ceremony in that they cleanse and release the grudges accrued over the deceased lifetime. As such, the 49-day ceremony follows a multi-religious amalgamation of Confucian and Korean shamanic traditions, which have been the foundation of Korean culture (Koo 2010, p. 267).

In the modern era of Japanese colonialism, Japanese Buddhism encroached on Korean Buddhism, and many Korean Buddhist rituals, including Korean Buddhist funerals, were once again vanquished (Han 2008, p. 141). Data on the use of the 49-day ceremony in the modern period are hard to come by, but one may speculate on its survival by studying Buddhist ritual books of the period. One such text, *Exemplary Examples for Buddhist Rituals* (작법귀감, 作法龜鑑), was published in 1826 by Baekpa Geungseon (백파긍선, 白坡亘璇, 1767–1852). A revision of Miscellaneous Ritual Texts (제반문, 諸般文), it contains detailed commentary on Buddhist rituals, serving as a comprehensive manual. It would become the basis for *The Manual of Essential Methods of Buddhism* (불교법요귀감, 佛敎法要龜鑑). However, *Exemplary Examples for Buddhist Rituals* was of particular importance as a model for *The Book of Buddhist Liturgies*, which is the basis for Korean Buddhist rituals today (Sørensen 1991, p. 167).

In 1931, Ahn Jin-ho compiled and published *Essential Compendium for Buddhists* (불자필람, 佛子必覽), which was an attempt to systematize and modernize Buddhist rituals performed by Koreans (McBride 2019, p. 84). At the time, the only ritual books used by temples were Joseon-era manuscripts, and even these were hard to come by. The *Essential Compendium for Buddhists*, a neatly arranged letterpress ritual book, was written not only in Hanja (the Korean writing system that makes use of Chinese characters) but also in Hangul (the distinct writing system in the Korean language) (Han 2008, pp. 142–43). This is a significant point, as Hanja was generally reserved for scholarly writing, which was inaccessible to the general population. Hangul, however, was more accessible as commoners could read it with relative ease. Thus, the revival of ritual texts written into Hangul and the level of interest that followed enabled Buddhist rituals, including the 49-day ceremony, to survive during the Japanese occupation of the peninsula.

After *Essential Compendium for Buddhists* sold out, it was revised in 1935 and widely circulated upon its publication. It was the only Buddhist ritual book available among Korean Buddhist communities until the 1960s (Han 2008, p. 147). *Essential Compendium for Buddhists* also served as a model for *Tongil beobyojip* (published in 1998), which inherited most of its content. This text is still used widely among Korean Buddhist temples to this day (Lee 2020, p. 278). Consequently, the procedures involved in the 49-day ceremony (see Section 3) will be explored by means of this most recent account, the *Tongil beobyojip*.

Although Buddhism and Buddhist rituals are now once again common in modern-day South Korea, the predominance of Confucian funerals during the Joseon Dynasty has carried over into modern times. According to a report from the 1970s, at that time, civilian funeral rites were held in the Confucian style of a three-year funeral (Koo 2008a, p. 220). However, according to the *Family Rites Ordinance* (가정의례준칙), enacted in 1976, this lengthy tradition was condensed into a 100-day ceremony. After the burial or cremation of the remains, the Confucian-based rituals were to be performed daily, at home, for 100 days in honor of the departed. In the last 10 to 20 years, however, this practice seems to have been largely abandoned. This decline in Confucian funerals may be due to the changing lifestyle of busy modern people (Koo 2008a, pp. 217–20). Due to data limitations, it is hard to obtain an accurate picture of how often the 49-day ceremony is practiced, but for those who feel burdened by daily funeral rituals, the 49-day ceremony is relatively short and is carried out seven times (over seven weeks) by monastics at a temple. It has been suggested

that the convenience of the 49-day ceremony is one of the main factors accounting for its growing favor ([Koo 2008b](#), p. 299).

Despite the 49-day ceremony being a Buddhist ritual, as of late, it has been utilized as a collective memorial ritual in the wake of major national disasters in Korea. This stems from the limited presence of established religious and traditional cultural funeral practices in Korea. Another possible factor is the perception among Koreans that the 49-day ceremony holds significance as both a Buddhist and a traditional Korean ritual ([Koo 2010](#), p. 255).

In the case of Protestantism in South Korea, the lack of religious grieving at funerals has become an issue ([Ahn 2013](#), p. 140). The Korean Catholic Church has not officially adopted Buddhist doctrine or rituals underlying the 49-day ceremony, but it does not prohibit the observance of such ceremonies. According to the wishes of the bereaved family, the church may commemorate the deceased through a 49-day mass ([Kim 2009](#)). This does not appear to be due to other religions adopting Buddhist rituals according to the 49-day ceremony but rather because the 49-day ceremony is recognized as a set of traditional and cultural rituals.

## 3. Procedures of the 49-Day Ceremony

If the bereaved family decides to hold a 49-day ceremony, they bring a photo of the departed to the temple rather than to their home after the cremation or burial of the body. The 49-day ceremony begins with a banhonjae (반혼재, 返魂齋), a ritual to bring spirits to the temple ([Jung 2018](#), p. 36). After the banhonjae, the first ceremony is held on the seventh day after the deceased's death. From that point forth the ceremony is repeated every seven days until a total of seven ceremonies have taken place. After this 49-day period, the spirit leaves this world, and the bereaved members of the family return to their daily lives ([Koo 2008a](#), p. 222). The most important of the seven ceremonies is the last one (on the 49th day) because the maximum time the dead can remain in an intermediary existence is 49 days. While the first six ceremonies hold special significance, the seventh and final ceremony is especially meaningful as it is an opportunity for the bereaved to bid a final farewell to the spirit of their loved one.

Traditionally, it has been required that the bereaved attend all seven of the ceremonies; however, this custom has become flexible as members of the congregation now tend to give precedence to other uses of their time and money. According to a study on the use of 49-day ceremonies, the seventh and final ritual was the most attended, as nearly half of the visitations took place during the final ceremony. Next, 20.8% visited during odd-numbered funerals (the first, third, fifth, or seventh), while 17.7% attended the first and last funeral. The main reason cited to explain why visitation was not held during all seven of the ceremonies is the burden of both the time and financial cost ([Koo 2008b](#), pp. 305–6).

The rituals of the 49-day ceremony are traditionally performed by Buddhist monks in temples. They are meant to be performed within the context of the three treasures (삼보, 三寶), namely, the Buddha, the dharma, and the sangha. In other words, to properly guide the spirit of the deceased, rituals are performed in a temple under the guidance of a monastic, who chants the sutras ([Koo 2009](#), p. 244). By reciting Buddhist teachings, the 49-day ceremony is a means of washing away the bad karma of the dead, presenting sacred energy with various mantras, and leading the spirit of the deceased to be reborn in the Pure Land (정토, 淨土) ([Koo 2009](#), pp. 248–49). The Pure Land, however, is not to be mistaken for the afterlife as expressed by the six realms of saṃsāra (육도윤회, 六道輪廻). The Buddhist view of the afterlife is that, based on one's karma, he or she is reincarnated endlessly in the six realms: heaven; human; jealous gods (asura); hungry ghosts; animals; and hell. The Pure Land is different from the heavenly realm of the six paths. When one is born in the Pure Land, one abides in nonretrogression (불퇴전, 不退轉), which is achieved through the attainment of patience based on the realization of the non-arising of phenomena (무생법인, 無生法忍). This is the state of never being born into the six realms again ([Han 2010](#), p. 63). The ultimate purpose of the 49-day ceremony is to bring the dead to the Pure Land, where they will no longer be reincarnated.

Although the procedures of the 49-day ceremony are divided into eight stages, according to the *Tongil beobyojip*, it has also been interpreted to be divided into five main phases. According to the five-phase explanation, the 49-day ceremony begins with receiving the deceased. After this, the bad karma which the deceased had accrued over his or her life is washed away. The third phase consists of prayers to the Buddha and bodhisattvas that the deceased may be reborn in the Pure Land. The fourth phase includes offering food to the deceased, which is then followed by a funeral ceremony and Buddhist teachings. Finally, the last step is to release the deceased (Yu 2018, p. 20). In Korea, when a person passes away, most people hold a three-day funeral, after which the deceased is buried or cremated. If, after the three-day funeral, the bereaved family decides to perform the 49-day ceremony, the rituals begin after the ritual to bring the spirit to the temple where it is enshrined.

The more specific eight stages, as expressed in the *Tongil beobyojip*, are as follows: (1) preparation (시련, 侍輦); (2) reception (대령, 對靈); (3) purification (관욕, 灌浴); (4) prayers for completion (신중작법, 神衆作法); (5) dharma teaching (설법의식, 說法儀式); (6) making offerings (불공, 佛供); (7) prayers for protection (중단퇴공, 中壇退供); and (8) performance of memorial rites (관음시식, 觀音施食) (DPBJO 2003, p. 21). Furthermore, although it does not appear in the table of contents of the *Tongil beobyojip*, the text indicates that there is an additional stage of relinquishing the spirit of the dead (봉송, 奉送). However, it is necessary to point out that several studies in Korea have included relinquishment as one of the stages of a 49-day ceremony, excluding dharma teachings or stage five (Yu 2018; Lee 2007; Chong 2021). The dharma teaching stage is a process in which the officiant delivers a sermon for the spirit of the departed and the bereaved family members who participated in the ritual. However, depending on the temple and the situation, sermons may not be given, and so many studies have excluded them from their model of the 49-day ceremony procedures (Koo 2008b, p. 312). Despite the variances among interpretations of the 49-day ceremony, recent studies have settled on the following eight phases, which this article, too, will abide by. Therefore, the following eight stages will be discussed below: (1) preparation; (2) reception; (3) purification; (4) prayers for completion; (5) making offerings; (6) prayers for protection; (7) performance of memorial rites; and (8) relinquishment.

The stage of preparation (시련, 侍輦) is composed of two characters, meaning to receive (시, 侍) and to lead (련, 輦), which refers to leading and receiving the spirit of the dead at the entrance of the temple. At this stage, the names of the guardian deities that protect the temple are invoked. These deities include the Śakra, the king of heaven (Indra), the four heavenly kings, and the eight groups of protector beings. An offering of tea is then made to the deities (DPBJO 2003, p. 271). This ritual is performed to introduce the deceased to these protector deities before the 49-day ceremony is performed. It is symbolic of the official entrance of the deceased into the temple (Lee 2020, pp. 279–83).

Next, the stage of reception (대령, 對靈) is composed of two characters, meaning to face or receive (대, 對) and spirit or ghost (령, 靈). This refers to the entrance of the deceased, past the guardians, into the temple. In this stage, food is offered to Amitābha (아미타불, 阿彌陀佛), who presides over the Pure Land. In addition, two bodhisattvas, Avalokiteśvara (관세음보살, 觀世音菩薩) and Mahāsthāmaprāpta (세지보살, 勢至菩薩), are called upon and worshiped. As Pure Land thought developed in Mahayana Buddhism, Amitābha played a major role in guiding sentient beings to be reincarnated in the Pure Land. It is stated in many sutras that a sentient being who chants Amitābha with a supreme mind will be reborn in the Pure Land (Koo 2011, p. 125). Therefore, Amitābha is treated as an important figure in Korean Buddhist funeral rites belonging to Mahayana Buddhism. Content related to Amitābha appears in various places in *Tongil beobyojip*'s 49-day ceremony ritual (DPBJO 2003, pp. 274, 304, 331, 343). In the context of the entire 49-day ceremony process, the reception stage is also meant to reassure and calm the spirit of the dead, leading them to comply with the procedures of the ritual (Jung 2018, p. 38).

The third stage, purification (관욕, 灌浴), refers to anointment (관, 灌) and bathing (욕, 浴) of the soul of the departed. Water is a symbol of purification, which is why some people

take a bath before performing certain sacred religious acts. In this step, toiletries and clothing for the spirit are prepared, and the symbolic act of bathing the spirit is performed (Oh 2019, p. 88). Part of this soul-cleansing ritual includes the recitation of mantras, including the bathing mantra (목욕진언, 沐浴眞言), the gargling mantra (수구진언, 漱口眞言), and the face-washing mantra (세수면진언, 洗手面眞言). These cleansing rituals specifically wash away the bad karma that has accumulated throughout the life of the deceased (DPBJO 2003, pp. 283–88).

The fourth stage consists of prayers for the safe completion of the ceremonies (신중작법, 神衆作法). This is when the ritual methods (작법, 作法) are performed in the presence of a group of guardian deities (신중, 神衆). According to the *Tongil beobyojip*, the content of this stage includes a reminder that the guardian deities fill the space around us (옹호성중만허공, 擁護聖衆滿虛空) and that the Buddha's words are to be believed in, accepted, and protected (신수불어상옹호, 信受佛語常擁護) (DPBJO 2003, p. 299). In this sense, the fourth stage invokes the guardian deities that protect the Buddha and his teachings and pray that the funeral rituals will be completed safely and without any obstacles (Lee 2007, p. 449).

The fifth stage is making offerings (불공, 佛供), including prayers, to the Buddhas and bodhisattvas. This is the stage where, through prayer, Buddhas and bodhisattvas bestow blessings on the deceased to aid their rebirth into the Pure Land. Especially in the 49-day ceremony, it is customary to offer prayers to Kṣitigarbha bodhisattva (지장보살, 地藏菩薩) (Koo 2011, p. 126). Sometimes referred to as the 'earth store' or 'earth womb', Kṣitigarbha bodhisattva vowed to save all people from the world of suffering, even entering hell to save the wicked. The *Tongil beobyojip* presents the invocation of Kṣitigarbha (지장청, 地藏請) as the main prayer, with Kṣitigarbha bodhisattva as the main object of that prayer (DPBJO 2003, p. 307). This seems to have been influenced by *The Sūtra of the Fundamental Vows of the Bodhisattva Kṣitigarbha* (지장보살본원경, 地藏菩薩本願經), as mentioned in Section 2, which states that if one performs good deeds for the dead for 49 days, the dead will be born in the Pure Land and receive great pleasure. Moreover, the benefits for the living family will also be great.

The sixth stage is prayers for protection (중단퇴공, 中壇退供). This is a ritual in which offerings (공, 供) made to the Buddhas and bodhisattvas are taken back (퇴, 退), and other offerings are made to the guardian deities who protect the Buddha and his teachings. In this ritual, the bereaved family and monks ask the guardian deities to protect the departed spirit so that his or her rebirth in the Pure Land will be safe (Jung 2018, p. 39).

The seventh stage is the performance of memorial rites (관음시식, 觀音施食). The first two characters (관음, 觀音) refer to the bodhisattva of compassion, Avalokiteśvara, while the latter two (시식, 施食) mean to bestow food. In this stage, food is prepared and offered to the deceased while sutras and mantras are recited to comfort them. This is meant to induce the deceased to realize the teachings of Buddhism and ensure their rebirth in the Pure Land (Choi 2019, p. 148). Along with Kṣitigarbha, Avalokiteśvara is a very important figure in the 49-day ceremony. One of the verses included in this stage emphasizes this with the following, 'I take refuge in Avalokiteśvara bodhisattva, whose great compassion relieves sentient beings in pain and difficulty' (나무 대자대비 구고구난 관세음보살, 南無 大慈大悲 救苦救難 觀世音菩薩) (DPBJO 2003, p. 336). As the verse suggests, Avalokiteśvara liberates sentient beings from suffering, and as such, the worship of Avalokiteśvara is very common in Korean Buddhist funeral ceremonies (Koo 2011, p. 128). Thus, the inclusion of Avalokiteśvara in the title of the seventh stage reflects Korean religious tradition and emphasizes the liberation of sentient beings. In Korea, Avalokiteśvara is widely believed to possess great powers of relief and to help sentient beings in all kinds of difficulties. In particular, the 49-day ceremony borrows heavily from the *Lotus Sutra* (법화경, 法華經), especially from the widely read chapter on The Universal Gate of the Bodhisattva Avalokiteśvara (관세음보살보문품, 觀世音菩薩普門品). Additionally, the Lotus Sutra frequently features the bodhisattva as a ritual object (Koo 2011, pp. 127–28). It is, therefore, speculated that Avalokiteśvara's devotional position is reflected in the 49-day ceremony.

In addition to this, during the first six stages, the officiating monk leads the funeral rites while standing in front of the bereaved. Beginning with the seventh stage, this monk performs the rituals from behind the bereaved. This is performed to give space to the family members who take turns prostrating before the photographs and memorial plaques of the deceased. During this time, the monk takes a secondary role, reciting the contents of the ritual book, while the bereaved take the lead. This enables the family to communicate with the deceased, thoroughly expressing their emotions (Koo 2008a, p. 237).

The eighth and final stage of the 49-day ceremony is relinquishment (봉송, 奉送), which means to respectfully (봉, 奉) send (송, 送) the spirit to the Pure Land. This stage serves as a farewell to the various Buddhas and bodhisattvas as well as the deceased. In addition, this section of the *Tongil beobyojip* contains the mantra for rebirth in the highest level of the highest stage of the Pure Land (상품상생진언, 上品上生眞言) (DPBJO 2003, p. 353). Here, the first two characters (상품, 上品) refer to the highest stage of the Pure Land, where mantra chanting functions like a prayer to ensure the spirit will be born in the best possible realm (Jung 2018, p. 40).

While each of the stages of the 49-day ceremony contain language specific to Korean Buddhism, each of these terms can be translated using more accessible terminology. For example, Chong (2021) suggests the first stage, siryeon, be expressed as the 'preparatory stage', and the final stage, bongsong, be expressed as 'sending' (p. 11). Table 1, below, summarizes the meaning of each stage in the 49-day ceremony process accordingly.

**Table 1.** Stages in the 49-day Ceremony.

| Stage: English | Stage: Hangeul and Hanja | Description |
| --- | --- | --- |
| Preparation | 시련 (侍輦) | Introducing the spirit of the deceased to the deities that guard the temple. |
| Reception | 대령 (對靈) | Receiving the spirit and serving it a simple meal. |
| Purification | 관욕 (灌浴) | Purifying the spirit of bad karma accumulated during the deceased's life. |
| Prayers for completion | 신중작법 (神衆作法) | Asking the guardian deities to help ensure the safe completion of the 49-day ceremony. |
| Making offerings | 불공 (佛供) | Making offerings to the Buddha and bodhisattvas and praying for the spirit's rebirth in the Pure Land. |
| Prayers for protection | 중단퇴공 (中壇退供) | Asking the guardian deities to protect the spirit and ensure rebirth in the Pure Land. |
| Performance of memorial rites | 관음시식 (觀音施食) | Offering food to the spirit, performing memorial rites, preaching Buddhist teachings, and guiding the spirit to the Pure Land. |
| Relinquish | 봉송 (奉送) | Letting the spirit go. |

The preceding has been a look at the history and details of the funeral rites known as the 49-day ceremony. Thus far, the focus has been on the deceased, and the rituals were practiced to ensure his or her arrival into the Pure Land. However, these ceremonies play another important role in the well-being of the bereaved. The next section is an investigation into how the rituals mentioned above serve the living.

## 4. Functions of the 49-Day Ceremony

As mentioned, the primary purpose and function of the 49-day ceremony is to ensure, through Buddhist funeral rites, that the deceased will be reborn in the Pure Land, a world described in the three Pure Land scriptures (정토삼부경, 淨土三部經), namely, in the Sutra on the Buddha Amitāyus (무량수명경, 無量壽命經), the Sutra on the Visualization of the Buddha Amitāyus (관무량수명경, 觀無量壽命經), and the *Amitâbha Sūtra* (아미타경, 阿彌陀經), as a place where there is no suffering or disease, and where one can live the



most comfortable and happy life (Han 2010, pp. 57–58). Through the 49-day ceremony rituals, the departed spirit is purified through various stages consisting of various mantras and Buddhist texts, offered food, and guided to the Pure Land through the power of the blessings of the Buddha, bodhisattvas, and guardian deities. However, these various procedures also serve as an important means of connection for surviving family members, not only as they take time to say goodbye to the departed but as an opportunity to cultivate meaningful relationships among themselves.

The 49 days leading up to the farewell of the departed can be of great comfort to the bereaved, as it gives them a chance to resolve unfinished issues (Lee 2007, p. 442). It is an opportunity for the bereaved to gain closure on unresolved feelings they may have about the deceased. The death of a loved one requires time and psychological healing to help those left behind accept and process their loss. In many cases, death leaves behind feelings of injustice and regret. Part of the function of a funeral ritual is to resolve these kinds of complex emotions by opening a space for dialogue among the living and dead. The 49-day ceremony is not the only way Koreans communicate with the dead, as shamanic rituals provide this opportunity, as do Confucian funerals, which have been a long-standing tradition in Korea. However, Korean shamanic rituals do not include funerals, i.e., rituals that take place immediately after someone's death, and Confucian funerals can last as long as 100 days (and in some cases up to three years), the inconvenience of which makes them rarely practiced by modern Koreans. The 49-day ceremony, however, takes place directly after death and for a relatively shorter period in a religious space (e.g., a Buddhist temple). In other words, a specific time and space are made available, which may be why these rituals are sometimes considered a more appropriate set of tools for mourning in the modern era. Moreover, lingering feelings of guilt can be eased if the individual in question feels he or she has carried out something to ease the pain of the spirit of the deceased. Helping a loved one pass into the Pure Land may, to some degree, make up for regrettable actions (or a lack of actions) while the deceased was alive. As Koo (2008a) points out, when a loved one dies, the bereaved benefits from rituals as they address psychological issues associated with grief, including guilt, regret, and loss, as well as the desire to send the spirit of the deceased to a higher realm (p. 299).

Each cultural tradition employs distinct mechanisms to confront the inevitability of death, with certain components potentially aligning more favorably with individual circumstances. The influence of modernization, exemplified by globalization and industrialization, has prompted adaptations in religious practices to accommodate the evolving lifestyles of individuals. Funeral rites in South Korea have not remained unaffected by these transformations.

It is necessary to note that the 49-day ceremony is not inherently superior to alternative funeral practices; rather, its appropriateness depends on specific contemporary conditions. One such condition is the necessity for connections, not only between the living and the deceased but also among those grieving. Additionally, the 49-day ceremony, as observed in public displays of mourning during national tragedies, has proven effective for individuals seeking solace and a platform to express their grief publicly, where their sentiments can reach politicians, policymakers, and others.

The primary function of the 49-day ceremony is linked to its secondary function, namely, the practice of rituals to cleanse the bad karma of the deceased, which consequently eases the difficult emotions of the bereaved. According to Jung (2018), the 49-day ceremony gives the bereaved who wish for the rebirth of the deceased into the Pure Land the strength to overcome their loss and other negative emotions as it helps them adjust to their new circumstances (p. 36). This is because the 49-day period after the death of a loved one allows for meetings between the deceased and the bereaved. This period allows the bereaved to deal with grief and provides a kind of foothold for accepting the death (Lee 2007, p. 446). Thus, the 49-day ceremony functions as a psychological and emotional buffer for bereaved families who have lost a loved one (Koo 2008a, p. 221).

It is not easy to cope with the pain of loss and make the necessary adjustments to return to functional life. There may be a strong desire to return to 'normal' life or life as it was before the loss. Thus, it is important that one has time to cope with and process their new circumstances. If repressed, various psychological problems are likely to emerge (Yu and Shon 2019, p. 126). Grief is a complex emotion consisting of various feelings related to loss. It may include intense sadness, depression, helplessness, loneliness, longing, self-blame, and discomfort and is a response to the emotional distress of deprivation (Zisook and Shuchter 1993, pp. 365–67). The Diagnostic and Statistical Manual of Mental Disorders (DSM-V) includes grief as a disorder. One can be diagnosed with such a disorder if one or more of the following symptoms persist for more than 12 months (in adults): persistent longing, intense sadness and emotional distress, and preoccupation with the deceased (Yu 2018, p. 14). Bereavement is a psychological state of holding on to the deceased, producing a range of negative emotions, including grief, anger, blame, and guilt, which can lead to emotional and social difficulties, as well as negative effects on physical health (Yu 2018, pp. 15–16).

In Yu's (2018) analysis, the process of grief includes three interrelated stages. The first involves understanding, accepting, and coping with the loss and the new situation it has created. The second step is to loosen one's attachment and perhaps even identification, which was lost. Finally, the third stage is to return to daily life (p. 17). While it is helpful to take time to work through these stages alone, sharing in grief with others and receiving their support is also required for the bereaved to have a thorough grieving process (pp. 3–4).

Several studies considering the relationship between the 49-day ceremony in South Korea and psychological healing have been conducted. One study included ten women (aged 40–60) who had lost a parent within the last year and who also practiced the ceremonies of the 49-day ceremony. Using consensual qualitative research methods, that study discussed the subjects' motivations for participating in these ceremonies, their experiences of grief over the 49 days, and their psychological state after completing the process (Yu 2018, pp. 24–28). Some of the psychological states reported by the participants included feeling sorry for the parent, feeling guilty that they were not a good enough daughter while the parent was alive, being grateful for the time they had with their parent, wishing the spirit of their parent a good rebirth, reliving memories with the parent, and sharing their grief with their family. Psychological states after completing the 49-day ceremony included feeling relaxed, comforted, relieved to have fulfilled their duty as a daughter, accepting the death of their parents, and reflecting on their own life and death (Yu 2018, pp. 31–51).

Another study considering grief experienced by bereaved families during the 49-day ceremony was conducted by Lim (2016). Participants in this study experienced separation from the deceased in two ways: those who lost a loved one suddenly, often due to unforeseen accidents; and those who lost a loved one to a foreseeable death, often due to a chronic illness and/or old age. Both mentioned feelings of guilt for not being the best person they could be for the deceased, sadness that the deceased had struggled in life, remorse and regret that they had somehow let the deceased down in life, and intense grief over the loss. That study also reported that their mourning experiences during participation in the 49-day ceremony included comfort in knowing that the deceased had been purified and that they had been sent to the Pure Land (Lim 2016, pp. 248–55).

It is worth noting that this study found that participants tried to communicate with the dead by having a conversation with them in their minds during the 49-day ceremony ritual (Lim 2016, pp. 256–57). This is in line with the theory that the 49-day ceremony acts as an occasion for meetings between the bereaved and the dead, which seems to help the bereaved release difficult emotions and let their grief unfold. That study further reported that the bereaved mentioned consciously attempting to detach themselves from the deceased, as they expressed the desire to move on from their grief and go back to the routines of their daily lives (Lim 2016, pp. 259–61). This corresponds to the three stages of grief mentioned

above: (1) acceptance and coping with the death; (2) detachment from the deceased; and (3) returning to normal life. Moreover, it provides a glimpse into the healing functions of the 49-day ceremony. While this qualitative study is limited due to its small sample size, it nonetheless supplies an important starting point for deeper discussion on this topic.

Finally, a study was conducted, which examined grief in those who held the 49-day ceremony after the death of a close family member (parent or sibling). A total of 20 Buddhists participated in a study where the researchers explored the participants' reasons for holding the 49-day ceremony and the changes in their emotional states. The reasons for performing the rituals included 'the family's religion has been Buddhist for generations' (six participants, or 30%), 'they wanted to do something for the deceased' (four participants, or 20%), 'they had interacted with a monk at the temple the deceased attended during their lifetime' (one participant, or 5%), 'for the deceased's rebirth into Pure Land' (6, or 30%), and 'they felt they were supposed to' (three participants, or 15%) (Chong 2021, pp. 13–23). In terms of psychological changes during the 49-day ceremony, participant reports were like those mentioned in previous studies. Some such statements include, 'My heart hurt, but the time spent praying for the deceased was meaningful', 'I felt sorry for the deceased', and 'I hope the deceased has been reborn in a good place'.

However, what the above-mentioned studies have not considered is an exploration of the functions and advantages of the 49-day ceremony within the context of large-scale, collective utilization. This article has highlighted the utilization of the 49-day ceremony in the aftermath of the Sewol ferry disaster (2014) and the Itaewon tragedy (2022). These instances have demonstrated the role of the 49-day ceremony in fostering healing among the populace, bringing together diverse groups of individuals to mourn, irrespective of their religious or familial affiliations.

While one of the primary purposes of the 49-day ceremony is to address and alleviate negative emotions, it is noteworthy that, in the context of national tragedies, this ceremonial practice has also encompassed the collective expression of anger. In the face of substantial losses on a large scale, this anger has frequently been directed toward societal institutions, fueled by a perceived sense of social or political injustice.

Examining cases such as the Sewol ferry disaster, resulting in the loss of hundreds of lives, and the Itaewon tragedy, where over 150 individuals perished in a crowd collapse, the public response has been characterized by outrage. This outrage is often channeled towards institutions, including local district offices, police departments, and the Ministry of the Interior and Safety, holding them culpable for failing to implement necessary preventative measures to avert such catastrophes. The nationwide observance of the 49-day ceremony in the aftermath of these events serves as a collective outlet for the expression of resentment and frustration, encapsulating a shared call for justice, appealing to individuals irrespective of their religious or non-religious affiliations.

The exploration of the 49-day ceremony in the context of national disasters as a vehicle for collective catharsis is a rich and multidimensional subject that beckons further scholarly attention. This research can not only deepen our understanding of the cultural and psychological dimensions of this ritual but also inform the development of culturally sensitive strategies for coping and healing in the aftermath of large-scale traumatic events.

## 5. Conclusions

This article has explored the history, procedures, and functions of the 49-day ceremony in Korean Buddhism, a set of rituals to guide the dead to the Pure Land, and a healing tool for the living. In Korea, these Buddhist funeral rituals have been passed down through the Goryeo and Joseon dynasties and continue to be used to this day. Since the 1970s, a shift has begun away from the more complex and time-consuming Confucian-style funerals toward the 49-day ceremony, presumably due to the changing lifestyle of modern people. The convenience of performing these rituals at a temple, as opposed to at one's home, is another reason for the shift. Moreover, after the death of former President Roh Moo-hyun, many Korean Buddhist orders held the 49-day ceremony in his honor. Many

people nationwide participated, including Buddhists and non-Buddhists, as well as media outlets, which further publicized the ceremony. Similar nationwide periods of mourning were held during the Sewol ferry disaster of 2014 and the Itaewon tragedy of 2022. Overall, during the last 50 years, there has been a renewed interest in these Buddhist funeral rituals in Korea.

This article has distinguished between two functions of the 49-day ceremony. The primary function is religious in nature, concerned with the afterlife of the deceased. Beginning with the preparatory stage, which introduces the dead to the deities who protect the temple, an eight-stage process is undergone where the spirit of the deceased is received, purified, prayed for, presented with offerings, and eventually released to the Pure Land. In addition to this is a secondary purpose, namely, functioning as a means of psychological healing for the bereaved. This is carried out in part by helping living members of the family to relinquish attachment to the deceased by playing an active role in sending the spirit of their loved one to the Pure Land. Furthermore, the psychological distress encountered in the face of loss, including sadness, depression, lethargy, and longing, requires time for the grief to be processed. Thus, the opportunity to meet with the dead for 49 days, and a kind of sacred grieving through religious ritual, helps the bereaved come to terms with their difficult emotions. Some have even argued that the 49-day ceremony should be adapted to focus more emphatically on the bereaved (as opposed to the deceased), supplying them with space to grieve and education on death. Such an adaptation would function not only as a send-off for the deceased but as a program providing resources for the grieving to learn about themselves and their feelings in times of loss (Chong 2021, p. 3). In particular, the seventh stage (performing memorial rites) is the portion of the 49-day ceremony that often involves the bereaved playing an active role in the ceremony. This would be an appropriate point to engage the bereaved in guided programs of reflection before the spirit is released. Furthermore, this would be a way of equipping the grieving with tools that they can then use at home in their daily lives. In other words, an adapted version of the ceremony could complement those who do not regularly attend temple services, enabling them to pray or meditate for the dead at home. Thus, while the framework of the existing method of the 49-day ceremony is effective, it could be even more so with some adaptations (Koo 2010, pp. 271–72).

Although the 49-day ceremony is a Buddhist tradition, it has also shown signs of helping those who do not identify as Buddhist. The 49-day ceremony has served as a healing tool on a social level in South Korea, as has been shown during the aftermath of several national tragedies. As such, the 49-day ceremony can provide us with some insights regarding the development of programs for those who have lost loved ones.

**Author Contributions:** Investigation, H.M. and B.D.S.; writing—original draft preparation, H.M.; writing—review and editing, H.M. and B.D.S. All authors have read and agreed to the published version of the manuscript.

**Funding:** This research was funded by the Ministry of Education of the Republic of Korea and the National Research Foundation of Korea, grant number NRF-2021S1A6A3A01097807.

**Data Availability Statement:** Not Applicable.

**Conflicts of Interest:** The authors declare no conflict of interest.

## Abbreviations

T = *Taishō Shinshū Daizōkyō*, 大正新脩大藏經 [Taishō edition of the Buddhist canon]. Ed. Takakasu Junjirō 高楠順次郎; et al. 100 vols. Tokyo: Taishō Issaikyō Kankōkai, 1924–1935.

## Notes

[1]    Other translations from previous studies include "49-day funeral ceremony" and "49 rites of Buddhism" as well as "sasipgu-jae" (Yu 2018; Oh 2019; http://www.koreanbuddhism.net/bbs/board.php?bo_table=4010&wr_id=5 (accessed on 17 July 2023)).

[2]　Apidamo Dapipasha Lun (阿毘達磨大毘婆沙論) T. 1545. vol. 27, p. 361b. "中有極多住 七七日. 四十九日定結生故".

[3]　Yuqie Shidi Lun (瑜伽師地論) T. 1579. vol. 30, p. 282a "又此中有 若未得生緣極七日住 有得生緣即不決定 若極七日未得生緣死而復生 極七日住 如是展轉未得生緣 乃至七七日住 自此己 後決得生緣".

[4]　The afterlife is not discussed in the Chinese scriptures with the same detail as it is in the Tibetan texts especially the *Tibetan Book of the Dead* (Lee 2017, pp. 25–26). However, a comparative study of the Tibetan scriptures would require a much deeper study, and the scope of this study could be extensive. For this reason, this article focuses on texts directly related to the Korean 49-day ceremony, primarily those in Chinese scripture.

[5]　Dizang Pusa Benyuan Jing (地藏菩薩本願經) T. 412. vol. 13, p. 784a "若能更爲身死之後 七七日內 廣造衆善 能使是諸衆生 永離惡趣 得生人天 受勝妙樂 現在眷屬 利益無量".

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
