# Peer review of "The Contents and Functions of the 49-Day Funeral Rites in Modern Korean Buddhism"

_religions, doi:10.3390/rel14121482_

Round 1

Reviewer 1 Report

Comments and Suggestions for Authors

Review of the academic paper titled “The Contents and Functions of the 49-Day Funeral Rites in Modern Korean Buddhism

Summary:

The paper explores the forty-nine-day Buddhist ceremony from two aspects and primary functions: sending off the deceased and providing spiritual and psychological comfort for the survivors. It outlines the textual origins of the rite, describes the procedure as it is practiced today and discusses its potential practical benefits in coping with loss.

General Comments:

The chosen topic is meaningful in terms of shedding light on the psychological and communal benefits of religions.

The paper is well structured with a clear argumentation except for the few points I point out below. It links the traditional past to the present skillfully. References to contemporary examples of performing the rite are particularly welcome, e.g., at the time of the Itaewon tragedy, Sewol ferry disaster etc. This gave the topic a topicality and relevance. Although the same examples are cited at least three times in the paper, which feels a bit superfluous.

A few points that could further improve the paper:

It would be meaningful to mention this or similar rites in India and/or China, especially if it influenced the Korean practice. It is not clear from the paper if the 49-jae is a rite that only exists in Korea, or there are some Indian or Chinese rites it is based upon. I think similar rites must exist in other countries as well. In other words, some historical and cultural contextualization would be useful regarding the origin of the rite, and how the Korean rite differs from similar rites in other countries.

Many studies and research are cited in the paper. Is might be good to emphasize what is the original aspect of this paper, that is, the contribution of the authors beyond summarizing the existing research.

Items in the bibliography is listed in English, although most of them are published in Korean. I think omitting the original Korean title of the papers and journals makes it hard for readers to identify and find these sources in case they want to investigate the topic further.

Also, I think it is enough to provide hangeul and hanja only after the first mention of a text or an expression, whereas the names of Buddhas and bodhisattvas need not be italicized.

Specific comments:

L 49-50. Firstly, the ceremony functions as a send-off for the deceased ensuring their place in the heavenly realm.

 At l107-108 it is stated that the “ritual is premised on the idea of reincarnation and is therefore centered on sending the dead to a better afterlife”, so the expression “heavenly realm” does not seem accurate here. The sentence should be rephrased.

L 59-61. Romanization should be consistent throughout the paper e.g., Tongilbeobyojip or Tongil-Bubyojip.

L 60. The Chinese characters for Tongilbeobyojip should be presented here.

L 63-65. In addition, the terms associated with 49-jae are originally written in Chinese characters, but since this article will focus on Korean literature, the Korean phonetic transcription will be used except for scripture names.

This sentence is a bit confusing, and it should be more precise. I think both hangeul and hanja were provided almost everywhere (which is useful), while in the names of Buddhist sūtras and translations from them, Chinese pronunciation was used (see the next comment).

L 78-80. Is there a specific reason, the Chinese terms are used in the translations? It seems to be because the translation is based on the Chinese translation of the sūtra, but I think it can be confusing for readers who cannot read Chinese. Since the paper discusses a Korean topic, I would rather use either the Sanskrit or the English translation in the main text, and supplement it with Chinese characters and pinyin or Korean pronunciation for important terms.

L 127-128. Veritable Records of the Joseon Dynasty

Original Korean name, hangeul, Chinese characters should be added. What kind of content was recorded in this text?

L 163. ‘Family Rites Ordinance’

Korean original with hangeul should be presented.

L 208-209. Hangeul is not necessary here since it has already been provided.

L 208-214. “As mentioned in section 2…”

This information would fit better in the section first introducing these books. It feels redundant here, just breaks the chain of thought.

L 221. “…leading the spirit of the deceased to be reborn in the Pure Land”

It would be useful to have some explanation about Buddhist afterlife, since three different versions were presented until now (and later): rebirth in the Pure Land, getting a place in a heavenly realm, and reincarnation (in the realms of humans, hells, hungry demons etc.). This might be confusing for someone who is not familiar with Buddhist concepts of afterlife. Where does this rite want to ensure rebirth? Later it seems that the aim is to ensure rebirth in the Pure Land, but the notion of Pure Land and heavenly realms seem to be blurred. It is probably blurred in Korean believers’ mind as well, but this academic paper could be more precise in this term, especially because it seems to be a differentiating factor from Confucianism.

L 228. “heavenly realm”

See above.

L 229. “…followed by a funeral ceremony…”

Some clarification is needed about what kind of funeral ceremony this is, because it was stated previously that the ritual takes place after the burial of the body.

L 283. Tongilbeobyojip should be in italics.

L 309. Sisik should be in italics.

L 319-321. The inclusion of ‘gwaneum’ in the title of the seventh stage, ‘gwaneumsisik’ reflects Korean religious features and the symbolism of a representative figure of liberating sentient beings.

So, this stage was not included in afterlife (especially 49-day) rituals in other countries? Being the acolyte of Amitābha in several scriptures, the worship Avalokiteśvara in funeral-related rites does not seem surprising to me. How is this a Korean feature? Does Avalokiteśvara have a special place in Korean religiosity? As I mentioned in the general comments, some historical and cultural contextualization would be useful regarding the origin of the rite, and how the Korean rite differs from similar rites in other countries.

L 352. Again, “heavenly realm” is not a good word choice here, since the Pure Land has nothing to do with heaven or heavenly realms in the saṃsāra.

L 353-354. “…the Jìngtǔ sānbù jīng (淨土三部經) Wúliángshòu jīng (無量 353 壽命經), Guān wúliàngshòu jīng (觀無量壽命經), and Āmítuó jīng (阿彌陀經)…”

It should be made clear with punctuation or rephrasing that the Wúliángshòu jīng, Guān wúliàngshòu jīng (觀無量壽命經), and Āmítuó jīng are actually the Jìngtǔ sānbù jīng. A translation of this term might be useful for this as well. Also, the description of the Pure Land could be more accurate. How is it different from the actual heavenly realms of Buddhism?

L 368-369. “Part of the function of a funeral ritual is to resolve these kinds of complex emotions by opening a space for a dialogue among the living and dead.”

This element of dialog is not that conspicuous for me in the 49-jae as it is in, for example, in Korean shamanic rituals. How this ritual gives more opportunity for addressing unresolved issues and emotions than other funeral rituals? Is it merely by the length and frequency of the rites? But, as mentioned earlier, Confucian rituals are lengthier. And musok has afterlife concepts that Confucianism might lack.

L 520. “…it must be free of Buddhist proselytizing or political objectives…”

Isn’t it free now? Sounds a bit critical me without giving any context or reason.

Author Response

First and foremost, the authors of this manuscript would like to sincerely thank reviewer one for their thoughtful and constructive feedback. The comments have been essential to the improvement of this article.

Responses to general comments:

  • The influences of Korean Buddhism outside of Korean (e.g. China) have been considered and included (117-159)
  • Please note that we have also included the Korean titles to the sources in the reference section.
  • With one exception when referring to Theravadin Buddhism (146-9), the technical terminology has been limited to hangeul and hanja.

Responses to specific comments:

  • The phrase “heavenly realm” has been removed throughout the article, frequently replaced with “Pure Land”.
  • Romanized terms have by and large been replaced with an English translation. In cases where they remain the reviewer’s suggestion was applied.
  • The reviewer’s point is well taken regarding the Chinese terms. The titles of texts as well as technical terms have been replaced with English translations with some rare exceptions when the phoneticized Korean was used.
  • An elaboration on the content that was recorded in the Veritable Records of the Joseon Dynasty (beginning with line 166).
  • The redundancy mentioned regarding lines 208-214 is well received. These lines have been removed.
  • The distinctions between various forms of afterlife has been well noted and adjusted by the authors (see lines 316-325).
  • Clarification is given about the kind of funeral ceremony (333-337).
  • The authors have expanded on the role of Avalokiteśvara in the Korean 49-day ceremony (428-435).
  • A section has been added to contrast the 49-day ceremony with Korean Shamanism and Confucian traditions (484-492).
  • Regarding the final lines of the article the reviewer’s comments have been concerned and changes made accordingly (635-641).

Reviewer 2 Report

Comments and Suggestions for Authors

This article analyses in detail a Buddhist meaning of Forty-Nine Day Ceremony (49-Jae) for the dead in Korea based on its historical changes and psychological functions. Given that there are only a few studies on this topic outside and inside Korean Buddhism, it seems very desirable and encouraging to cover this important topic in an English journal. In this sense, I would like to make a few suggestions and recommendations to make this manuscript more appealing and inspirational to English readers interested in Korean Buddhism:

1. The 49-Jae is a tradition developed in Mahayana Buddhism, if so, it needs to explain why and how. The authors are engaged in explaining its historical development only in comparison with (Korean) Confucianism, especially the concept of reincarnation, but it is not enough to understand the specific ritual in a more diverse context.  

2. There were and are a variety of funeral rituals in all societies, interacting with each other in different contexts and for different reasons. If so, the 49-Jae was and is subject to the influence of folk religions, shamanism, or ancestor worship in the past or contemporary Korea. It does not mean that the 49-jae is out of a Buddhist tradition originally. But a relationship the 49-Jae and other religious traditions should be included in this sort of discussion at least briefly. It seems a more balanced perspective.

3. Regarding a psychological function of the 49-Jae, the authors focus simply on an individual dimension of deprivation and compensation. But social or collective dimension is also found very often in Korean society, as mentioned on the Itaewon incident or Sewol Ferry tragedy in the article. In this sense, the authors need to add some analysis of such collective memory related to the 49-Jae ritual. 

Comments on the Quality of English Language

It is necessary to revise the manuscript in accordance with a more formal style of writing in English. Please take care of an exact usage of comma, colons, semi-colons, and space. I would like to suggest a more intensive proofreading made by a native-speaker. Thank you

Author Response

First, we would like to express our thanks to reviewer two for taking the time to read this article and give the constructive comments that have helped to improve our work.

  • As the reviewer rightly points out there is a need to expand on the development of the 49-day ceremony not just outside of Korean Buddhism. The influences of Korean Buddhism outside of Korean (e.g. China and Early Buddhism) have been considered and included throughout particularly in lines 117-159.
  • A brief discussion on Korean Shamanism and ancestor worship in Confucianism in the context of the 49-day ceremony. Please see lines 212-225 & 484-492.
  • Aside from the use of the 49-day ceremony during national tragedies the social/collective dimension of the 49-day ceremony has been further considered and an expanded throughout. Please see lines 273-286.

Furthermore, this article was proofread by a native English speaker with a more formal style of writing in mind. Grammar has been adjusted where necessary and furthermore, Romanised terminology has been replaced with English translations to make for a more fluent reading experience.

Reviewer 3 Report

Comments and Suggestions for Authors

The primary contribution of this essay is its description of the 49-day Buddhist funeral ceremony in contemporary Korean Buddhism. It doesn’t advance any really new information, although it does make available in English a generally thorough introduction to the Buddhist funeral process in the present-day Republic of Korea. This paper shows some promise but it is not publishable in its present form.

This reviewer has attached a pdf with some suggested corrections.

The primary shortcomings of this essay are that the author does not really know who his/her audience is and the author does not know the accepted norms of bibliographic citation for either Korean Studies or Buddhist Studies. If the audience is Western scholars of ritual and/or Western scholars of Buddhism, the author needs to use English terms more consistently to demonstrate that he/she truly understands the material. This is because the technical terms that the author presents in Chinese or Korean, should be presented in English translation, with the Chinese and Korean transliterations placed in parentheses. Tone marks for Chinese are not necessary. There is nothing special about using either Chinese or Korean terms for standard Buddhist concepts. Make it easier on the reader and provide an appropriate English translation.

The author asserts that the 49-day ceremony has had a long history in Korea. Based on the authority of the modern Tongil beobyojip (1998 [using 2003 printing]), the author claims that there are records of the 49-day ceremony being practiced during the Unified Silla and Goryeo periods (see lines 111-117). However, to this reviewer’s knowledge, there is no clear evidence that the 49-day ceremony was practiced in early and medieval Korea. Neither of the terms, sasipgujae 四十九齋 or chilchiljae 七七齋 (or chilchiril 七七日), is found in the Samguk yusa, Samguk sagi, Goryeosa, Goryeosa jeoryo, and in Buddhist epigraphy from the Unified Silla and Goryeo periods. Since the author claims this essay to be a “history” (lines 4, 73) of 49-day funeral rites in Korea, he/she should provide at least a few examples from the Unified Silla and Goryeo periods.  In his Joseon Bulgyo tongsa 朝鮮佛敎通史 (1918), Yi Neunghwa 李能和 (1869-1945) asserted that 49-day ceremonies have continued in Korea since the beginning of the Joseon period; but the evidence supporting his position is also unclear.

The Samguk yusa does contain some stories of hell experiences, such as “Seonyul Comes Back to Life” 善律還生 (see Samguk yusa 5, T 2039, 49.1013c24-1014a16). On this see, Na Hee La 나희라. 2008. Tongil Silla wa Namal Ryeochogi jiok gwannyŏm ui jeon’gae 통일신라와 나말려초기 지옥관념의 전개, Han’guk munhwa 韓國文化 43: 245-265.

In addition, the author should refer to an essay published in English on death, burial, and the afterlife in medieval Korean Buddhism: Vermeersch, Sem. 2014. Death and Burial in Medieval Korea: The Buddhist Legacy. In Death, Mourning, and the Afterlife in Korea:  From Ancient to Contemporary Times, ed. Charlotte Horlyck and Michael J. Pettid, 21-56. Honolulu:  University of Hawai‘i Press. [Dr. Vermeersch does not find evidence of 49-day ceremonies in the Goryeo period, although he notes that rebirth within 49 days was a standard Buddhist doctrine.]

The Bulja pillam 佛子必覽 is not a “compilation of rituals that had been handed down throughout history” lines 140-141). Rather, it was the first attempt to systematize and modernize Buddhist rituals performed by Korean Buddhism. See McBride, Richard D., II. 2019. Must Read Texts for Buddhists and the Modernization of Korean Buddhist Ritual. Journal of Korean Religions 10.1: 83–122.

In addition, before the Bulja pillam, there was at least one Korean Buddhist ritual manual that was published from woodblocks:  Geungseon’s 亘璇 (1767–1862) Jakbeop gwigam 作法龜鑑 (Exemplary Examples for Buddhist Rituals), which was published in 1827; see Han’guk Bulgyo jeonseo 韓國佛敎全書, vol. 10, 552b–609b. This ritual manual even has a section titled: “Proper Rites for Guiding the Spirit” or “Correct Rite for Summoning Spirits” (daeryeong jeongui 對靈正儀), see HBJ 10:560b–562b. The Jakbeop gwigam was itself a revision of Jeban mun 諸般文 (Miscellaneous Ritual Texts), a large ritual manual published in woodblock form in 1610 and which was republished with some changes in different editions in the seventeenth and eighteenth centuries.  See Sørensen, Henrik H. 1991-1992. A Bibliographical Survey of Buddhist Ritual Texts from Korea. Cahiers d’Extrême Asie 6: 159-200 (see esp. pp. 167-168).

The names of Indian gods, buddhas, and bodhisattvas should not be italicized because, as names, they are proper nouns (see lines 251, 259–267, 293-294, 309, 312-318)

line 45 (passim): 49-jae should be something like 49-day ceremony

line 60 (passim): Tongilbeobyojip should be Tongil beobyojip

line 61:  Tongil-Bubyojip should be Tongil beobyojip

line 61 (passim): Seongmunuibeom should Seongmun uibeom

line 78-79: instead of zhongyou, the author should use “intermediate existence” (jungyu, Ch. zhongyou 中有; Skt. antarā-bhava)

line 81:  instead of shengyou, the author should use “birth existence” (saengyu, Ch. shengyou 生有; Skt. upapatti-bhava; the period between conception and birth)

line 82: instead of siyou, the author should use “death existence” (sayu, Ch. siyou 死有; Skt. maraa-bhava)

line 82: instead of benyou, the author should use “basic existence” (bonyu, Ch. benyou 本有; Skt. pūrva-kāla-bhava; the period between birth and death)

line 93:  instead of zhongyou, the author should use “intermediate existence”

line 101: instead of equ, the author should use “unfortunate destinies” or “unwholesome rebirths”

line 119: instead of sungyueokbul, the author should use “promoting Confucianism and rejecting Buddhism” (sungyu eokbul 崇儒抑佛)

line 127:  Veritable Records of the Joseon Dynasty should be Joseon wangjo sillok 朝鮮王朝實錄 (Veritable Records of the Joseon Dynasty)

line 140:  Buljapillam should be Bulja pillam

lines 154, 209, 211: Seogmunuibeom should be Seongmun uibeom

line 216: sambo or the three treasures should be the three treasures (sambo 三寶)

line 221: use Pure Land (jeongto 淨土)

lines 231–234: regarding the terms: siryeon 侍輦, daeryeong 對靈, gwanyok 灌浴, sinjungjakbeop 神衆作法 (use sinjung jakbeop 神衆作法), seolbeobuisik 說法儀式 (use seolbeop uisik 說法儀式), bulgong 佛供, jungdantoegong 中壇退供 (use jungdan toegong 中壇退供), gwaneumsisik 觀音施食 (Gwaneum sisik 觀音施食), the author should introduce and use English translations of these terms.

line 271: does not mean to “provide water”; it means “to anoint” or “to consecrate”

line 276: mogyokjineon should be mogyok jineon (mantra for bathing the spirit/soul)

line 276: sugujineon should be sugu jineon (mantra for rinsing out the mouth of the spirit/soul)

line 277: sesumyeonjineon should be sesumyeon jineon (mantra for watching the face and hands of the spirit/soul)

line 295: jijangcheong should be “invocation of Kṣitigarbha”

line 333: mantra for the highest of the stages of birth in the Pure Land should be “mantra for rebirth in the highest level of the highest stage of the Pure Land” (sangpum sangsaeng jineon 上品上生眞言).

However, why not follow the style of providing the English translation for the mantra first in lines 275-277?

Regarding the table on pp. 7-8, the author’s rendering of the “stages” does not correspond to what is described in the preceding paragraphs. If the eight stages are so important, why does the author simplify them to six stages in the table?

Line 352: “heavenly realm, such as the Pure Land”. According to Buddhist doctrine, the Pure Land is not a “heavenly realm.” By definition it is precisely not a heavenly realm because one is not a deity if one is reborn there. The goal of all Korean Buddhist funerary rituals is rebirth in the Pure Land. Rebirth in the Pure Land is the most general and most common goal in East Asian Mahāyāna Buddhism, but is should not be mistakenly equated to rebirth in a heavenly realm.

The information in most of the references is incorrect and misleading. Most of the secondary sources used are actually in Korean; however, the way they are presented suggests that they are in English. Furthermore, many basic conventions, such as italicizing journal titles and book titles has been ignored. This whole section needs to be revised.

(Lim 2016) should be:

Lim Hae-young 임해영. 2016. 49 jaee chamyeohan yujokdeul-ui aedo gyeongheom yeon’gu 49재에 참여한 유족들의 애도 경험 연구 [A Study on the Mourning Experience of the Bereaved Family who participated in 49-Day Funeral Commemorations]. Han’gukhak 한국학 39.2: 237–271.

(Lim 2021) should be:

Lim In-young 임인영. 2021. Sangjwabu Bulgyo-ui jesa uisik yeon’gu 상좌부불교의 제사의식 연구 [A Study on the Funeral in Theravada Buddhism]. Dongasia Bulgyo munhwa 동아시아불교문화 47: 129-154.

Line 584:  Joong-ang sangha University should be Joong-Ang Sangha University

Line 587: In the reference to Zisook and Schuchter, the title of the article is missing from The Journal of Clinical Psychiatry, which should be italicized.

Comments on the Quality of English Language

Author Response

First, a sincere thank you to reviewer three as he/she provided the authors with valuable insights regarding the improvement of this manuscript. The thorough reading and comments are much appreciated and have been looked at carefully. Please note the responses to your inquires as follows.

  • The format of the bibliography has been adjusted with particular emphasis given to sources written in Korean. Furthermore, throughout the article the titles of texts as well as technical terms that had been Romanized have now been translated into English (with a few exceptions).
  • Regarding the history of the 49-day ceremony the authors have expanded the article (lines 117-176). We are grateful to the reviewer’s suggestions, which have included some valuable resources (i.e., Vermeersch 2014, McBride 2019) which have been very helpful not only to the authors’ understanding of the history of the 49-day ceremony but have made for a more well-rounded article.
  • With reference to sources provided by the reviewer adjustments have been made to more suitably describe the Bulja pillam (230-237).
  • Proper names that had been Italicized have now been edited.
  • The reviewer gives a detailed list of terms that need editing. These suggestions have all been considered and the corresponding changes made. Most Romanized terms have been translated into English. Some significant changes were made to the description of the 8 stages of the 49-day ceremony (338-342; 353-355; 456).
  • Regarding the use of the term “heavenly realm” the authors have decided to replace this term with “Pure Land” and furthermore gave a brief explanation, which can be found in lines 316-325.

Round 2

Reviewer 2 Report

Comments and Suggestions for Authors

Basically, the revised version looks publishable, reflecting faithfully what reviewers suggested. But I think it still has to make clear the function of the 49-day ceremony in the following aspects:

First, the authors need to explain more on why the Buddhist 49-day ceremony is preferred by non-Buddhists to Christian, Shamanistic, or Confucian memorial ceremony, as put in line 470 below. 

Second, I do not think that a long, but simple, description of previous studies, as put in line 557 below, makes authors' arguments of psychological effects more appealing. The authors' new findings of psychological effects should be added to those previous research results. 

Once this kind of correction is done, I think this article can be published. Thank you.

Comments on the Quality of English Language

Without any necessary reasons, it is better to delete all Korean letters in this article, including references. It will make a higher level of readability.

Author Response

The authors of this manuscript would once again like to thank reviewer 2 for their thoughts and comments. As in round 1, round 2 has included valuable insights that have improved the quality of this work.

Please note the changes made throughout the manuscript especially lines 493-505, which address the first concern. On lines 588-613 the reviewer’s second concern is addressed.

Regarding the use of Korean characters, while the inclusion of Korean letters (and Korean Romanization in the referencing section) may take away from the overall readability of the article, given that it is an article on Korean Buddhist practice, and considering that not all readers may not be familiar with Chinese characters, we think it is best to keep the Korean characters in the text. We agree that the format of the Korean characters and Romanization in the reference section is cumbersome however, it has been formatted that way as a result of comments from round 1 reviews.

Reviewer 3 Report

Comments and Suggestions for Authors

This revision is a great improvement. The author has followed most of the recommendations this reviewer made in his review of the earlier version of the manuscript. It is publishable with some minor corrections.

Line 81:  seven·seven => seven times seven

Line 128: (Vermeersch p. 46) => (Vermeersch 2014, p. 46)

Line 141: (nimitta) =>  (nimitta)

Line 143: (cui cita) => (cui citta)

Lines 159-161: Sources from the Unified Silla and Goryeo periods that contain either of the terms 49-day ceremony or seven-seven-day ceremony are scare. => There are no Korean sources from the Unified Silla and Goryeo periods that contain either of the terms 49-day ceremony or seven-seven-day ceremony.

Line 178: sungyu eokbul => sungyu eokbul

Line 205: ‘mu’ => mu

Line 277: 49-jae => 49-day ceremony

Line 350: jeseokcheon, the king of heaven => Śakra, the king of heaven (Indra)

Line 351: sacheonwang or the four heavenly kings => the four heavenly kings

Line 352: palbujung or the eight kinds of beings => the eight groups of protector beings

The traditional list of the (palbu, Ch. babu 八部) is heavenly dragons, yakas, asuras, garuas, kinnaras, mahoragas, humans, and non-humans.

Line 422-423: Avalokiteśvara Bodhisattva and the Universal Gate => “The Universal Gate of the Bodhisattva Avaloktiesvara” (chapter titles not italicized)

Line 453: Three Principle Texts of the Pure Land Tradition => three Pure Land scriptures

Line 454: Muryangsu Gyeong => Sutra on the Buddha Amitāyus.

Line 454-455: Guan Muryangsu Gyeong => Sutra on the Visualization of the Buddha Amitāyus

Line 513: In Yu (2018)’s analysis => In Yu’s analysis (2018)

References

The author needs to provide bibliographic citations for primary sources, including:

T: Taishō shinshū dai zōkyō 大正新修大藏經 [Taishō edition of the Buddhist canon]. Ed. Takakasu Junjirō 高楠順次郎, et al. 100 vols. Tokyo: Taishō Issaikyō Kankōkai, 1924–1935.

The three Buddhist scriptural texts (lines 629-631) need bibliographic citations that should include Taisho number, volume number and page numbers.

Apidamo dapiposha lun

Yuqie shidi lun

Dizang pusa benyuan jing

Journal titles and book titles need to be italicized.  See the attached pdf with title words that need to be italicized underlined in green. Something needs to be italicized in nearly all entries.

Entry no. 10 (lines 649-651) is incorrect. The name of the journal is Bulgyo hakbo.

Entry no. 11 and entry no. 21 are both from 정토학연구, however, the author is inconsistent in his/her transliteration of the title:  Jeongto-hak yeon’gu or Jeongtohak yeon’gu

Bibliographic entries no. 5, 12, and 16 do not provide the proper information.  If the source is a newspaper, then the bibliographical information needs to be provided following a standard citation style. If it is a website, then the bibliographic information for that website needs to be provided.

Comments on the Quality of English Language

Author Response

We would like to thank the reviewer once again for his/her very thorough and helpful notes and suggestions. The reviewer’s careful attention has allowed us to improve our article with efficiency.

Please note that we have taken all of reviewer’s notes into consideration and made edits accordingly with one exception: Reference number 10 (ie. Sukmoonyibum vs. Seongmun uibeom). Although accurate, we have decided not to adjust the romanization as suggested by the reviewer because it conflicts with the romanization as given by the author. For the original article please see the following link.

https://academic.naver.com/article.naver?doc_id=927863529